# Dental Decision-Making in Pediatric Dentistry: A Cross-Sectional Case-Based Questionnaire Among Dentists in Germany

**DOI:** 10.3390/medicina60111907

**Published:** 2024-11-20

**Authors:** Bakr A. Rashid, Ahmad Al Masri, Christian H. Splieth, Mustafa Abdalla, Julian Schmoeckel

**Affiliations:** 1Department of Pediatric Dentistry, University Medicine Greifswald, Walther-Rathenau Str. 42a, 17475 Greifswald, Germany; bakr.rashid@stud.uni-greifswald.de (B.A.R.); ahmad.almasri@uni-greifswald.de (A.A.M.); splieth@uni-greifswald.de (C.H.S.); 2Department of Orthodontics, University Medicine Greifswald, Walther-Rathenau Str. 42a, 17475 Greifswald, Germany; 3MyPediaClinic, Dubai Healthcare City, Dubai P.O. Box 505206, United Arab Emirates; drmustafaabdalla@hotmail.com

**Keywords:** dentistry, primary teeth, dental treatment, decision-making, nitrous oxide sedation, general anesthesia

## Abstract

*Background and Objectives*: The most recent guidelines and recommendations regarding treatments of dental caries in children are shifting towards evidence-based minimal or non-invasive approaches aiming to preserve the vitality of teeth and potentially reduce the need for dental general anesthesia. This study investigated the treatment recommendations of dentists actively practicing pediatric dentistry in Germany regarding different patient cases with caries in primary teeth. *Materials and Methods:* The questionnaire was distributed on paper or online to pediatric dentists and general dentists practicing pediatric dentistry. Five cases of children with dental treatment needs representing a variety of clinical situations were selected for the questionnaire. Considering four different scenarios regarding pain symptoms (yes/no) and cooperation level (good/low) for each case resulted in 20 questions, where the preferred treatment option could be chosen out of 21 options ranging from observation only to extraction with/without different sedation techniques. The answers were categorized into three categories for each case and scenario according to guidelines, recent scientific evidence, and recommendations (recommended, acceptable, or not recommended/contraindicated). *Results*: In total, 222 participants responded to the survey (161 female; 72.5%). In 55.2% of the total 4440 answers, the participants chose a “recommended” treatment option, in 16.4% “acceptable”, but in 28.4%, a “not recommended” treatment, which ranged for the five cases between 18.7 and 36.1%. While pain and low cooperation levels led to more invasive and justified treatment choices (only 26.3% “not recommended”), less severe scenarios resulted more often in “not recommended” options (pain with good cooperation: 31.0%; or low cooperation without pain: 32.6%). The dentist’s age, experience, and educational background did not significantly correlate to choosing “not recommended” treatment options. *Conclusions:* A child’s pain and cooperation level greatly impact the treatment decisions made by dentists, with a risk of too invasive treatment options in low-severity cases. Substantial disparities in treatment recommendations for caries in primary teeth persist among dental practitioners regardless of their age, experience, and educational background.

## 1. Introduction

Decision-making in medicine and dentistry is one of the most critical steps before initiating any clinical treatment. Especially in pediatric dentistry, many factors should be considered in the process of decision-making, far beyond tooth level, such as the cooperation level of the child, the overall oral treatment need, caries risk and activity, the accuracy of the diagnosis, potential risks of the treatment and sedation options, the financial aspects, and the duration of the treatment [1,2,3]. In modern dentistry, reaching a treatment decision is not done by the dentist alone (anymore). A change from the traditional, rather so-called “paternalistic” approach to more participative approaches with informed consent has occurred in medicine and dentistry in the past decades [4], which is oriented toward the autonomy of the patients and includes their competencies. Pediatric health care has the challenge of having “immature” patients with limited cooperation and understanding of health issues, as well as difficulties regarding parental preferences. This makes decision-making in pediatric dentistry far more challenging than other specialties in dentistry, which should not be underestimated, as the initial recommendations of the dentist for a certain treatment path lay the basis for later communication and decision-making with informed consent from the parent or the guardian [3].

The challenges in decision-making in pediatric dentistry extend beyond finding the balance between parental preferences and children’s capabilities, as the treatment options on the tooth level have witnessed a shift towards minimal or non-invasive approaches. The conventional restoration (CR) of a cavity on a primary tooth using amalgam, glass ionomer cement, resin composite, or polyacrylic acid-modified composites used to be the standard treatment in pediatric dentistry [5]. These types of restorations are still widely used, even though they show weak performance with a high need for retreatment [6] and may be associated with a high risk of restoration failure and pulpal complications [7]. With a correct pulp diagnosis of reversible pulpitis, selective carious tissue removal or caries sealing (e.g., the Hall technique) could be performed, avoiding complete caries removal near the pulp and minimizing the risk of pulp exposure, which can preserve the vitality of the primary molar [8]. Non-restorative cavity control (NRCC) is also a modern concept that aims to arrest the progression of caries through oral hygiene after removing overhanging enamel and dentin, if needed, to make the lesion accessible, which should be followed by repeated and regular biofilm removal as well as fluoride application to stimulate the remineralization of the carious tooth structure and stop the activity of the lesion arresting its progression, which requires the high adherence of patients and/or parents to manage the lesion [2]. The integration of silver diamine fluoride (SDF) and the Hall technique (HT) into modern caries management in pediatric dentistry has expanded the dental treatment spectrum in pediatric dentistry and provided options for children with limited cooperation, offering effective solutions with minimal invasiveness. These evidence-based approaches provide comprehensive care while prioritizing the preservation of tooth structure and patient comfort [9,10,11]. Moreover, sedation methods such as nitrous oxide or general anesthesia (GA) widen the spectrum of treatment approaches even further, which makes the decision on the best treatment approach in different circumstances in pediatric dentistry even more challenging. In general, no single treatment option is the only possible, and no treatment option can always be considered the best. However, some good treatment options might be considered unsatisfactory in certain circumstances or situations. For instance, a pulpotomy on a symptomless primary tooth with (deep) proximal dentin caries is an evidence-based approach with reported high success rates. Still, in a case of low cooperation, the treatment would probably require sedation or even GA. Considering the risks of GA along with other less invasive evidence-based approaches, such as the HT with a diagnosis of reversible pulpitis, a pulpotomy could be considered disadvantageous as the HT is possible even in the case of limited cooperation, avoiding the need for DGA [12].

With these wide ranges of treatment options and approaches in pediatric dentistry and the shift towards minimal and non-invasive treatments, it remains unclear whether daily clinical practices are following the updates in the literature. Therefore, this study aimed to investigate the treatment recommendations of dentists actively practicing pediatric dentistry in Germany based on different cases of caries in primary dentition, with suggested scenarios considering pain symptoms and cooperation levels.

## 2. Materials and Methods

This questionnaire-based study includes five cases with clinical and/or radiographical pictures suggesting different dental treatment options with different scenarios.

### 2.1. Study Population and Data Collection

As the study aimed to investigate the different approaches of dentists in different clinical scenarios and cases in pediatric dentistry, it was necessary to include a large number of participants actively practicing pediatric dentistry, regardless of their qualifications and specialization. The questionnaire was sent via email to all listed pediatric dental clinics from the website of the German Society for Pediatric Dentistry (DGKiZ, N = 360) and to all participants in, as well as graduates of, a postgraduate master’s program in Preventive and Pediatric Dentistry at the University of Greifswald in Germany (N = 108). Moreover, the questionnaire was handed out personally to general dentists in Germany (N = 86) who were attending continuing education courses in the field of pediatric dentistry before the start of the course to avoid bias. To calculate the sample size, the population size of dentists practicing pediatric dentistry was anticipated to be around 3500. With a 95% confidence interval and a 10% margin of error, a sample size of 94 was needed. Data collection was done in the period from March 2020 to August 2022.

### 2.2. The Questionnaire

A questionnaire was designed to include five typical cases of dental caries in primary teeth with clinical and/or radiographical pictures. Participants were asked to choose their recommended treatment option for each case 4 times for different scenarios that reflect real-life scenarios in pediatric dentistry. The four scenarios for each case were as follows:
No reported pain, cooperative child(P−, ↑);No reported pain, uncooperative child(P−, ↓);Reported pain symptoms, cooperative child(P+, ↑);Reported pain symptoms, uncooperative child(P+, ↓).

The predefined treatment options for the cases with the different scenarios in the survey were 20 in total. They ranged from observation to invasive treatment options such as endodontic treatment or extraction of the tooth, considering also the treatment setting, such as nitrous oxide sedation or GA. The participants had an extra space after each question for typing a treatment option other than the listed ones. Traditional treatment options such as complete caries removal and filling and modern, less traditional treatment modalities such as silver diamine fluoride (SDF) and the Hall technique (HT) were all included as options. An overview of the treatment options and the cases can be seen in Table 1.

The five selected cases included a range of clinical situations concerning children with dental treatment needs, which covered most of the problems and challenges in caries management in primary teeth (Table 1):**Case 1:** arrested deep carious lesions (ICDAS 6) on first primary molars (74, 84) in a preschool child;**Case 2:** active deep proximal carious lesion (ICDAS 5) on a second primary molar (85) in a preschool child with high caries experience;**Case 3:** proximal carious lesion on a first primary molar (74, likely ICDAS 4, detected in bitewing radiograph as D2 lesion) in an elementary schoolchild;**Case 4:** clinical and periapical X-ray showing a proximal ICDAS 5 lesion and an occlusal ICDAS 4 lesion on a second primary molar with a dentine bridge between the lesion and the pulp in a preschool child;**Case 5:** clinical picture of semi-active dentine carious lesions on anterior primary teeth (Early Childhood Caries; 3-year-old child).

All the cases and questions focused on a specific tooth, and not on all carious teeth in the pictures because patient-level decisions are much more complicated and require a thorough explanation for many related factors, which would lengthen the questionnaire and the time required to fill it out and thus reduce the response rate. However, patient-related aspects were still considered in the scenarios mentioned above, mainly regarding pain symptoms and the cooperation level of the child. Other relevant information on the patient level is obtainable from the given pictures/radiographs, such as caries experience/risk/activity and phase of dentition.

Moreover, to investigate the factors influencing dental decision-making, the demographic data of the participants were collected, as well as answers to further questions regarding the number of children treated per week, years of experience as a dentist, years of experience with nitrous oxide sedation and GA, and years of practice as a specialist in pediatric dentistry.

### 2.3. Ethical Considerations

This study was conducted in full conformance with the principles of the “Declaration of Helsinki” and Good Clinical Practice (GCP) and within the laws and regulations of Greifswald University. Whether on the paper form or online, informed consent was obtained from all participants before filling out the questionnaire. Ethical approval was obtained from the Ethical Committee of the University of Greifswald (BB 052/20). The clinical intra-oral pictures of the patients were presented anonymously to study participants, as consent was taken from the parents/guardians of the children to use the intra-oral photos and/or X-rays in anonymous form for research purposes.

### 2.4. Treatment Recommendations

Three specialists in pediatric dentistry from the University of Greifswald met to develop an outline for rating the treatment options for the different cases with consideration of the scenarios; two of the three were specialist pediatric dentists and members of the university teaching staff with clinical experience, while the third was a last year postgraduate student in a master program in Preventive and Pediatric Dentistry. A fourth specialist pediatric dentist, who was the head of the Department of Preventive and Pediatric Dentistry at the university and of the mentioned master’s program, with a leading role in many worldwide recognized organizations in the field of cariology and pediatric dentistry, was consulted for his opinion in the treatment recommendations. The decision on categorizing the treatment options was thus based on the actual international recommendations and guidelines, yet with consideration of clinical real-life experience. The treatment options were divided according to the case and the suggested scenario into the following categories:❖**Recommended treatment options:** treatment options that follow the up-to-date evidence-based recommendations and guidelines for the management of the mentioned tooth/teeth, considering the suggested scenario regarding pain symptoms and the cooperation level of the child;❖**Possible and acceptable treatment options:** treatment options that do not follow the up-to-date evidence-based recommendations and guidelines for the management of the mentioned tooth/teeth, considering the suggested scenario regarding pain symptoms and the cooperation level of the child. However, although these treatment options are not up-to-date, they are still not contraindicated and could be performed;❖**Not recommended and contraindicated treatment options:** treatment options that do not follow the up-to-date evidence-based recommendations and guidelines for managing the mentioned tooth/teeth, considering the suggested scenario regarding pain symptoms and the cooperation level of the child, which are disadvantageous or even contraindicated.

### 2.5. Statistical Analysis

The questionnaire responses were exported to an Excel sheet (Microsoft Office 2021), anonymously coded, assigned to the categories of recommendation level as in Table 1, and exported into IBM SPSS for Windows (Version 23.0) for analysis. Frequencies and percentages were calculated for all qualitative variables, while means and standard deviations (SDs) were computed for quantitative variables. To investigate the factors influencing dental decision-making, the number of chosen “not recommended and contraindicated” treatment options was analyzed as the dependent variable. The independent variables considered were age, years of experience as a dentist, experience with nitrous oxide (in years), experience with general anesthesia (in years), years of practicing as a specialist, and the number of children treated per week.

A negative binomial regression model was used to investigate the factors that might lead to choosing an increased number of contraindicated treatment options. A further analysis was performed for each of the independent variables separately to investigate if any of these variables would show an effect on the total numbers of selected “not recommended/contraindicated” treatment options, where a Mann–Whitney test was performed for the categorial variables and Pearson correlation was used for the continuous variables. Moreover, a chi-square analysis was performed to analyze the differences in the distribution of the answers to the three categories of recommendation level between the cases and the scenarios. Statistical significance was set at a *p*-value of <0.05.

## 3. Results

In total, 222 participants responded to the survey. A total of 54 responses were in paper form (24.3%), while the majority of responses were collected through the online version (75.7%, *n* = 168). The total number of participants consisted of participants who are still attending or have completed a postgraduate master’s program in Pediatric and Preventive Dentistry in Germany, general dental practitioners in Germany, and members of the German Society for Pediatric Dentistry (DGKiZ e.V.). As efforts were made to invite as many practitioners as possible to participate in the survey, participants were asked to forward the online version of the questionnaire to colleagues in the same field. Thus, the precise number of participants who received the invitation and the exact response rate could not be calculated.

Table 1 shows the five cases of the questionnaire with the suggested scenarios and all the treatment options that were available in the questionnaire, as well as the percentages of the answers of the participants for each scenario (↑: cooperative child, ↓: uncooperative Child, P+: pain symptoms, P−: no pain symptoms) categorized into the three above-mentioned groups.

Table 2 provides the demographic and biographic characteristics of the respondents, along with the results of the negative binomial regression to investigate the variables’ relation to the total number of not recommended/contraindicated treatment options for each participant.

The Pearson correlation and Mann–Whitney tests revealed no significant association between choosing the “not recommended/contraindicated” treatment options with any of the study variables, such as years of experience, the educational background of the dentist in pediatric dentistry, the number of patients per week, etc. (Table 2).

The total of answers in the not recommended category in the five cases ranged from 18.7 to 36.1% according to the cases, regardless of the specific scenarios, as illustrated in Figure 1. Case 4 (cavitated D2 approximal lesion) showed the highest percentage of chosen not recommended treatment options (36.1%), while case 1 (inactive lesions) had the lowest percentage with 18.7%. The differences between the cases in the distribution of answers to the assigned categories were statistically significant, as shown by the chi-square independent test (*p* < 0.01).

When considering the specific scenarios, regardless of the cases, there were statistically significant differences between the scenarios in the distribution of the answers, as depicted in Figure 2 (chi-square test *p* < 0.01). The percentages of the chosen not recommended treatment options for the scenarios “no pain and uncooperative patient” and “pain and cooperative patient” (32.6% and 31.0%, respectively) were higher than for the scenarios “no pain and cooperative patient” and “pain and uncooperative patient” (23.9% and 26.3% respectively).

The percentage of the chosen not recommended/contraindicated treatment options did not differ when considering only the cooperation level of the child regardless of the pain symptoms. In contrast, the percentage of the chosen recommended treatment options was much less in scenarios with limited cooperation (Figure 3). These differences were statistically significant, as shown by the chi-square independent test (*p* < 0.01). The same pattern was also observed when considering pain symptoms regardless of the cooperation level, where pain symptoms lead to fewer chosen recommended treatment options (Figure 4). The differences between the scenarios “pain” and “no pain” in the distribution of answers to the assigned categories were statistically significant, as shown by the chi-square independent test (*p* < 0.01).

Among the treatment options as seen in Table 1 that were classified in the category “not recommended and contraindicated” for specific scenarios, there were eight treatment options that were selected by more than 30% of the participants in these specific scenarios, which are further summarized and explained in Table 3.

## 4. Discussion

This study aimed to explore and analyze decision-making processes among dental professionals in the field of pediatric dentistry. Understanding the methods employed and the characteristics of the study sample is crucial for interpreting the findings and assessing the generalizability of the results. To the authors’ knowledge, this is the first research in Germany to explore dental decision-making in pediatric dentistry, which also includes various samples with different clinical backgrounds. The hypothetical clinical case scenarios can measure opinions about the care that might be provided in one particular case [21]. These scenarios and cases are not a tool for measuring the actual care a dentist would provide, as this will also be subjective to other factors that are not enclosed in a scenario, such as parental wishes, financial aspects, and the availability of treatment options in practice.

This study is a questionnaire-based survey focused on self-reporting. Such methods have gained more popularity in recent years in medical research [22], as they can give more understanding of real-life clinical daily practice. In order to recruit as many participants as possible, the questionnaire was available in both paper-based and online formats, which allowed for a varied participant pool, considering the varying preferences and convenience of potential respondents [23]. The total number of respondents (*n* = 222) offers a generous dataset for analyzing decision-making trends in pediatric dentistry, where the majority of respondents are females, reflecting the gender distribution among young dentists in Germany [24], especially in pediatric dentistry, with the majority of the members of the DGKiZ e.V. being females [25]. A deeper look at the demographical background of the study population shows different backgrounds regarding the years and places of undergraduate dental education. This means that the base of dental knowledge of the participants is from different schools and concepts, which might lead to variability in decision-making [26]. A great variety was observed in all the other demographic characteristics of the study population, such as years of experience and the number of children treated per week, etc. This variety enriches the study’s external validity, capturing a range of perspectives from individuals at different stages of their dental careers, which makes the study population as near as possible to the real-life situation of practicing dental professionals, in order to be more representative of the target group as recommended in such surveys [27]. The study population’s variety and heterogenicity were reflected in the experience in sedation techniques in pediatric dentistry, where 66.2% of the respondents had experience with GA compared to only 46.8% with experience in nitrous oxide. This might be the reason behind many participants also choosing invasive GA in cases of good cooperation to treat acute pain and in cases of no cooperation and no pain symptoms instead of choosing less invasive treatment options, probably with the aid of nitrous oxide sedation, which could reduce the need for dental GA in children [28]. A possible explanation for the higher number of participants with experience in GA compared to nitrous oxide is the fact that the National Health Insurance in Germany would cover the costs of GA sedation if justified by a lack of cooperation and the presence of dental treatment need for children under the age of 12 [29], while the patient would individually cover the costs of nitrous oxide sedation.

Overall, most of the answers in all cases with different scenarios were within the recommended and acceptable treatment spectrum. Still, seven out of the twenty scenarios of the cases had a total of not recommended or contraindicated treatment options of more than 30%. Three of these seven cases were cases 3, 4, and 5 with a scenario of lack of cooperation and no pain symptoms, where the dentists tend to recommend the GA. However, the caries could be inactivated and arrested, with a great chance of healing the pulp due to the absence of pain symptoms. The case with the highest percentage of not recommended and contraindicated answers (61.3%) was the case of Early Childhood Caries with pain symptoms and good cooperation, where the recommendation should be towards teeth extraction or endodontical treatment, probably chairside or under nitrous oxide sedation, rather than fillings or strip crowns under GA. This might reflect the pressure to achieve esthetics in the area of anterior teeth in young children, as parents are more willing to accept non-esthetic treatment options for their children in posterior teeth than in the upper anterior teeth [30]. Besides the treatment modality, the choice of the restoration type was in many of the cases towards dental fillings, either using compomer or composites, which is similar to another questionnaire study in Saudi Arabia where 73.8% of the dentists also reported the use of composite fillings on carious primary teeth [31]. However, there are clear recommendations to use prefabricated crowns rather than fillings, especially in multi-surfaced lesions in primary molars [9,13,32], due to the clear, better longevity of the crowns, with less failure over a longer period of time [6].

As our results clearly show, pain symptoms and the cooperation of the child greatly influence the treatment options and recommendations of the dentists, potentially leading to choices that may not align with the guidelines and recommendations. In our results, the scenarios with limited cooperation clearly had a higher percentage of not recommended answers and a lower percentage of recommended answers, highlighting the influence of the cooperation level of the child on the decision-making process. While the lack of cooperation is mostly linked to dental fear, a study in Japan was able to prove that cooperation and dental fear do not have a causal relationship [33], necessitating the need for behavioral management concepts and techniques in pediatric dentistry to avoid the need to treat every uncooperative child under GA [1]. Signs of dental fear and anxiety can be predicted at the very first appointment [34], resulting in planning suitable steps for stepwise treatments, which in turn can prevent the failure of treatment due to a lack of cooperation. Almost analogous to the influence of cooperation levels, the presence of pain symptoms resulted in fewer recommended answers, showing also an influence on decision-making. However, it must be noted that pain response is individualized and influenced by various factors inherent to the procedural context [35]. Moreover, child-reported pain symptoms, especially in small children, are not valid and clearly defined. In our study, we used the terms “pain”, “spontaneous pain”, and “sensitivity”, which are descriptive of different states of clinical situations resulting in a different diagnosis. It cannot be excluded that the participants of this study did not quite differentiate between these terms, which would influence the decision-making process, as many practitioners in pediatric dentistry might not put a value on the patient-reported symptoms and concentrate mainly on clinical and radiographical examinations to reach their decisions, due to the limitations of self-reporting in children. However, pain and cooperation do influence each other, as sudden pain during treatments in children might cause a lack of cooperation in future sessions [36]. The dentists are, therefore, encouraged to minimize the experience of pain and discomfort during pediatric dental treatment, e.g., with the administration of local anesthesia [35,37]. The elimination of procedural pain would allow for a better judgment of symptomatic pain of dental origin.

Considering the differences between the cases regardless of the different suggested scenarios, it was observed that case 4, showing a deep ICDAS 5 approximal carious lesion on a primary molar, had the highest percentage of not recommended or contraindicated answers (32.9%). In contrast, case 3, showing an ICDAS 4 approximal carious lesion on a primary molar, had the lowest percentage of recommended answers (35.7%). This reflects the great challenges and difficulties practitioners face in diagnosing and treating approximal carious lesions in children and highlights the need for further education and training in this area. Restorative treatment thresholds based on radiographic lesion depth also vary substantially among dentists on permanent teeth in adults [38], where the problems of cooperation and pain are not relevant, as in pediatric patients. Our results could not detect true patterns or risk factors leading to an increased number of not recommended or contraindicated answers on an individual level, which might be due to the great heterogenicity of the study population. However, this heterogenicity reflects the actual diversity among practitioners actively practicing pediatric dentistry. Therefore, it cannot be concluded that more experienced dentists make better decisions or that younger practitioners are aware of the actual recommendations. Dental practitioners are thus encouraged to always seek further education to keep up-to-date with the latest updates and recommendations based on recent studies.

Our results showed a great variation in decision-making among pediatric dentists, which may be due to several factors. The most obvious explanation could be the great variations in the study population’s educational/clinical background [26] and work experience [39]. The differences and variations in decision-making should not be considered disadvantageous in general, as it is also noticeable in other dental specialties such as restorative dentistry [40]. Most commonly, dentists gain most of their knowledge in their undergraduate studies and adopt treatment scenarios and experiences of their own through the years, which become very challenging to change over time due to multiple barriers, such as the lack of knowledge and fear of making mistakes [41,42]. However, when it comes to making decisions for treatments using sedation techniques that carry high risks of complications such as GA [17], clinicians must be well aware of other alternatives such as nitrous oxide sedation and the possibility of non- or minimally invasive management options for dental caries when no pulpal involvement (such as indicated by pain symptoms) is clear, even with limited cooperation due to very young age. Nevertheless, these results should be interpreted with caution, as decision-making is a very complex process where many factors play a role and interact with each other, and due to the difficulties defining a gold standard with “correct” and “wrong” answers. In the planning of this study, efforts were made to build a categorization that would represent the most up-to-date recommendations and guidelines and to include the aspect of clinical experience through a team of multiple experienced specialized pediatric dentists with leading roles in teaching pediatric dentistry on the national and international levels. To minimize bias in the assessments of the answers, the treatment options for each case and scenario were categorized before the data analysis and are shown transparently in Table 1.

As in any other research, our study has several limitations that should be considered when interpreting the results. Most important is the generalizability and representativity of the study sample. Although the data collection process considered all potential practicing pediatric dentists, this specialty in Germany is not exclusively for specialists and could be practiced by general practitioners. Although all the members of the DGKiZ listed on their website were contacted (*n* = 360), only 11.4% (*n* = 41) of them responded by completing the survey. Moreover, the total number of DGKiZ members is nearly 1900, meaning the members listed on the website represent less than 20% of the members. Therefore, our results can give an idea about the variation and differences in decision-making among a sample of practicing pediatric dentists but does not represent all pediatric dentists in Germany. Another limitation of the study is the design of the questions on the tooth level, with the consideration of pain and cooperation on the patient level, while the real-life approach would have been to design the questions showing the whole dentition and adding questions regarding every tooth, to make more realistic decisions, especially considering GA. On the other hand, designing the questionnaire in this manner would lengthen the questionnaire and probably decrease the response rate dramatically, making the analysis and the results even more complex. Nevertheless, this study also has strengths that need to be discussed; the inclusion of pain and cooperation in such a questionnaire is an addition to many similar studies, and it showed a great influence on the decision-making. Moreover, the heterogenicity of the study population gives a closer look into the variety of the community in general and especially among practicing dentists, including experienced as well as freshly graduated dentists, who are either committed to pediatric dentistry and treat more patients per week or who only partially treat children, besides performing regular dental treatments on adults.

## 5. Conclusions

A great variety in dental decision-making was observed among practicing pediatric dentists in Germany. Pain symptoms and cooperation levels influence the decisions of the practitioner, with an especial risk of too-invasive treatment options in low-severity cases. A notable portion of the practitioners may suggest treatment options that might be considered not recommended or even contraindicated, highlighting the need for adhering to evidence-based dental guidelines and recommendations in daily practice and the need for continuous education over the years.

## Figures and Tables

**Figure 1 medicina-60-01907-f001:**
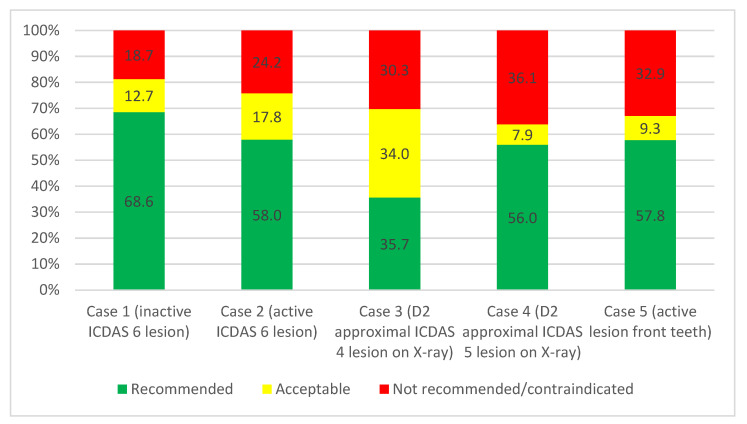
The percentages of the treatment options in the different cases were categorized according to the level of recommendation (*n* = 888 answers for each case in the four different scenarios from the 222 participants), differentiated by recommended, acceptable, and not recommended/contraindicated.

**Figure 2 medicina-60-01907-f002:**
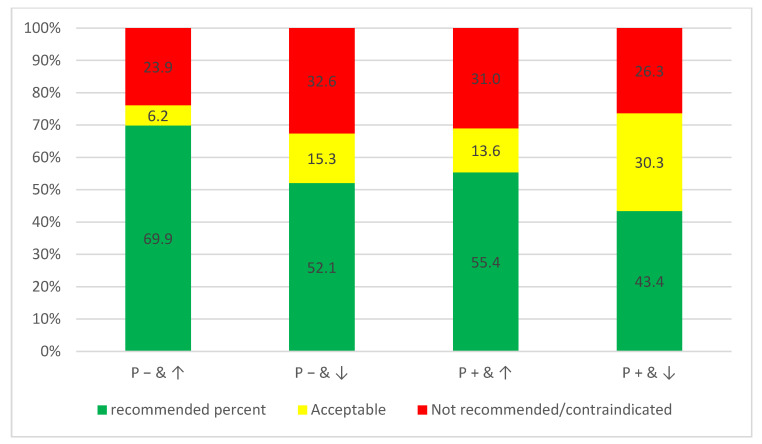
The categorization of the chosen treatment options regarding the suggested scenarios from the cases pooled together and shown as percentages (*n* = 888 answers for each scenario from all cases for the 222 participants). Differentiated by recommended, acceptable, and not recommended/contraindicated for ↑: cooperative child, ↓: uncooperative Child, P+: pain symptoms, and P−: no pain symptoms.

**Figure 3 medicina-60-01907-f003:**
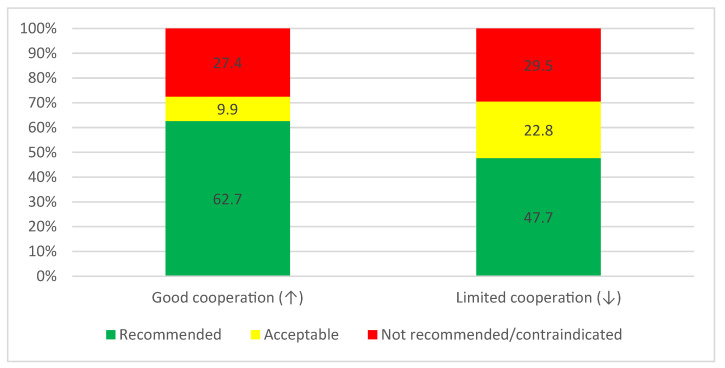
The categorization of the chosen treatment options regarding the cooperation of the patient from the cases pooled together, shown in percentages (*n* = 888 answers for each scenario from all cases for the 222 participants). ↑: cooperative child, ↓: uncooperative child.

**Figure 4 medicina-60-01907-f004:**
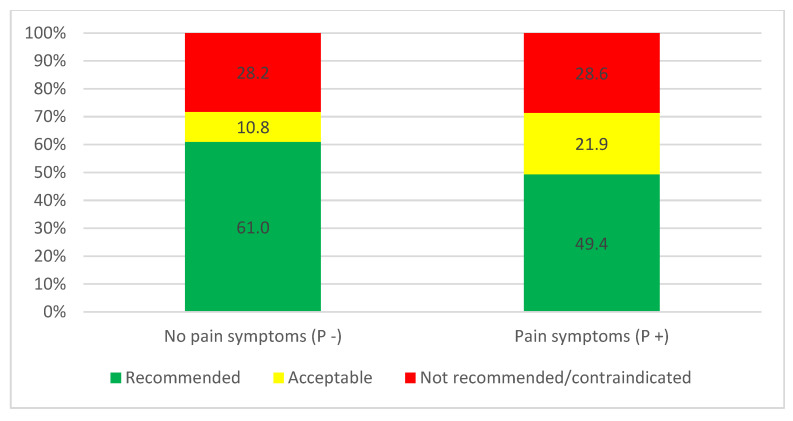
The categorization of the chosen treatment options regarding the symptoms of the patient from the cases pooled together, shown in percentages (*n* = 888 answers for each scenario from all cases for the 222 participants). P+: pain symptoms, P−: no pain symptoms.

**Table 1 medicina-60-01907-t001:** The cases and scenarios along with the percentage-wise assignment of each treatment option to the categories of level of recommendation (colors), and the results of the participants in percentages (*n* = 222).

Cases	Case 1Arrested ICDAS 6 on #74 and #84	Case 2Active ICDAS 5 #85	Case 3ICDAS 4 Proximal#74	Case 4ICDAS 4 Okklusal and ICDAS 5 Proximal #65	Case 5ECC#52–62
Clinical/radiographical pictures	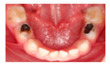	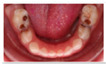	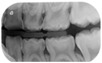	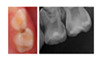	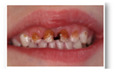
Treatment options/different scenarios *	P− ↑	P− ↓	P+ ↑	P+ ↓	P− ↑	P− ↓	P+ ↑	P+ ↓	P− ↑	P− ↓	P+ ↑	P+ ↓	P− ↑	P− ↓	P+ ↑	P+ ↓	P− ↑	P− ↓	P+ ↑	P+ ↓
No treatment/observation only	2.7	6.3	-	1.4	-	3.2	-	1.8	0.9	4.5	0.5	2.7	0.9	1.8	0.5	2.7	0.9	4.5	2.7	3.6
Non-restorative caries control: fluoride varnish and brushing instructions	10.8	30.6	-	0.5	1.8	8.6	-	2.7	2.7	18.0	0.9	6.3	2.7	10.4	0.5	1.8	15.3	24.8	1.4	3.2
Silver diamine fluoride application	2.3	19.4	0.5	5.0	1.4	24.8	0.5	4.1	1.4	14.0	0.9	8.6	0.9	20.7	1.4	4.1	17.1	34.2	6.3	7.2
Atraumatic restorative treatment with glass ionomer filling (ART)	3.6	4.5	2.7	3.6	1.8	9.0	2.3	6.8	1.8	3.6	2.3	2.7	2.3	9.5	2.3	2.7	3.2	4.1	1.8	0.9
GIC filling (with complete caries removal)	2.7	1.4	1.8	1.4	5.9	2.7	4.1	2.3	4.5	3.2	3.2	1.8	3.6	3.2	1.8	2.3	0.5	0.5	-	0.9
Compomer filling (selective caries removal)	15.3	2.7	6.8	1.4	19.4	2.3	4.5	1.4	23.0	5.9	14.0	2.7	15.8	3.6	6.8	0.5	11.7	1.8	7.2	0.5
Compomer filling (complete caries removal)	8.1	-	4.5	-	17.1	0.9	8.6	0.5	33.8	8.1	19.8	5.4	25.2	3.6	6.3	0.5	3.3	0.5	2.3	-
Zirconia pediatric crown	0.5	-	0.9	-	-	-	-	-	-	-	-	-	-	-	-	0.5	9.9	5.4	5.9	3.6
Strip crown composite restoration	-	-	-	-	-	-	-	-	0.5	-	0.5	-	-	-	-	-	28.8	6.3	10.4	2.3
Stainless steel crown (SSC) in Hall technique (no caries removal, no preparation)	24.8	13.5	1.4	1.8	13.5	15.3	2.7	3.6	12.2	19.8	6.8	11.7	16.2	14.0	1.4	0.5	0.5	0.5	0.5	0.9
SSC in conventional technique (complete caries removal, preparation)	8.6	0.9	1.8	0.5	10.8	1.8	6.8	0.9	7.7	1.4	5.9	0.9	6.8	1.8	5.9	0.9	--	-	0.5	-
Pulpotomy and SSC	11.7	1.4	37.4	1.4	18.9	0.9	36.9	-	7.7	-	26.1	2.3	19.4	0.9	39.6	2.3	-	-	1.8	-
Pulpotomy and SSC with nitrous oxide sedation	4.5	3.6	6.8	15.8	4.5	7.7	8.6	19.4	1.4	9.5	8.1	16.7	3.2	11.3	9.0	21.2	0.5	-	0.9	-
Pulpotomy and SSC under general anesthesia	0.5	8.1	3.6	19.4	2.7	17.6	2.7	25.2	0.5	8.1	1.8	20.3	0.9	14.9	1.8	23.9	0.5	0.9	1.8	2.7
Calcium hydroxide/iodoform paste, pulpectomy, and SSC	1.4	-	8.6	-	1.4	0.5	10.4	0.5	0.9	0.5	3.6	*-*	1.4	0.5	9.5	0.9	-	0.5	4.5	-
Calcium hydroxide/iodoform paste, pulpectomy, and SSC with nitrous oxide sedation	-	0.9	4.1	5.0	-	0.5	2.7	6.3	-	0.5	0.9	4.1	0.5	1.4	4.5	7.7	-	0.5	0.9	4.5
Calcium hydroxide/iodoform paste, pulpectomy, and SSC under general anesthesia	0.5	0.9	0.5	7.7	0.5	2.3	1.8	7.7	1.4	3.2	1.4	4.5	-	1.8	1.4	9.5	0.9	1.4	3.6	7.2
Local anesthesia extraction	1.8	-	12.2	0.5	-	0.5	3.6	0.9	-	-	1.8	-	-	-	5.4	0.9	2.3	0.9	16.7	2.3
Local anesthesia extraction with nitrous oxide sedation	0.5	1.4	5.9	9.5	0.5	0.5	3.2	5.0	-	-	1.4	5.0	-	-	2.3	6.8	-	2.3	14.0	7.7
Extraction under general anesthesia	-	4.5	0.9	25.7	-	1.4	0.9	11.3	-	-	0.5	4.5	0.5	0.9	-	10.8	5.5	11.3	17.1	52.7
**Percentage of chosen recommended treatment options**	71.6	74.3	74.8	52.7	59.5	57.7	65.3	49.1	76.6	19.8	34.2	11.7	58.1	45.0	70.3	50.9	83.8	63.5	30.6	52.7
**Percentage of chosen possible and acceptable treatment options**	13.1	3.6	5.0	30.2	1.8	25.2	7.7	36.0	13.1	41.4	48.2	33.8	-	-	-	31.1	3.2	6.3	8.1	19.8
**Percentage of chosen not recommended or contraindicated treatment options**	15.3	22.1	20.3	17.1	38.7	17.1	27.0	14.9	10.4	38.7	17.6	54.5	41.9	55	29.7	18	13.1	30.2	61.3	27.5

Green colored: Recommended treatment. Yellow colored: Acceptable treatment. Red colored: Not recommended/contraindicated treatment. * Abbreviations and symbols: ↑: cooperative child, ↓: uncooperative Child, P+: pain symptoms, P−: no pain symptoms, ART: atraumatic restorative treatment, ECC: Early Childhood Caries, GIC: glass ionomer cement, ICDAS: International Caries Detection and Assessment System, SSC: stainless steel crown.

**Table 2 medicina-60-01907-t002:** The demographic characteristics of the study population and the correlation of these variables with the number of not recommended/contraindicated answers.

	Mean Value ±SD	*p*-Value (Negative Binomial Regression as Continuous Variable)	Groups	Total Number	*p*-Value (Negative Binomial Regression as Categorial Variable)
**Age**	37.1 ± 9.8	0.582	-	-	-
**Sex**	-	-	Male	61 (27.5%)	0.704
			Female	161 (72.5%)	
**Children treated per week**	45.7 ± 78.2	0.402	-	-	-
**Experience as a dentist (years)**	11.7 ± 9.1	0.972	-	-	-
**Experience with nitrous oxide sedation (years)**	2.4 ± 4.9	0.403	Yes	104 (46.8%)	0.999
			No	118 (53.2%)	
**Experience with GA (years)**	4.89 ± 7.1	0.768	Yes	147 (66.2%)	0.534
			No	75 (33.8%)	
**Experience as specialist pediatric dentist (years)**	3.03 ± 5.3	0.649	Yes	120 (54.1%)	0.341
			No	102 (45.9%)	

**Table 3 medicina-60-01907-t003:** The treatment options, contraindicated or not recommended in specific scenarios, were chosen in ≥10% of these specific scenarios by participants, with the explanation for grouping them as contraindicated.

Treatment Option and Scenario	“Not Recommended/Contraindicated” Answers in %	Reason of Contraindication
Case 2 (Scenario: P− and ↑) Compomer filling (complete caries removal) in deep cavity	17.1%	Although symptomless, complete caries excavation in deep carious lesions is not recommended due to the risk of pulp exposure [13,14,15].
Case 2 (Scenario: P− and ↑) SSC in conventional technique (complete caries removal, preparation) in deep cavity	10.8%	Although symptomless, complete caries excavation in deep carious lesions is not recommended due to the risk of pulp exposure [13,14,15].
Case 3 (Scenario: P+ and ↓) Pulpotomy and SSC (caries media) under general anesthesia	20.3%	In this case, the pain was described as sensitivity on trigger, which indicates reversible pulpitis. GA carries risks of major and minor complications and should be avoided if possible [16,17]. There is no indication for GA as the carious lesion is not deep and, despite low cooperation, can be treated with minimal invasive options that do not require cooperation, such as the Hall technique [9,18].
Case 4 (Scenario: P− and ↑) Treatment of active caries approximal and occlusal ICDAS 4 with compomer filling (complete caries removal)	25.2%	Although symptomless, complete caries excavation in deep proximal carious lesions is not recommended due to the risk of pulp exposure [13,14,15].
Case 4 (Scenario: P− and ↓) Treatment of active caries approximal and occlusal ICDAS 4 with pulpotomy and SSC under general anesthesia	14.9%	GA carries risks of major and minor complications and should be avoided if possible [16,17]. There is no indication for GA as the carious lesion is not deep and, despite low cooperation, can be treated with minimal invasive options that do not require cooperation, such as the HT [9,18].
Case 5 (Scenario: P− and ↓) EEC in a 3-year-old with primary maxillary anterior teeth with active caries, extraction under general anesthesia	11.3%	GA carries risks of major and minor complications and should be avoided if possible [16,17]. There is no indication for GA as the carious lesion is not deep and, despite low cooperation, can be treated with minimal invasive options that do not require cooperation, such as the SDF [9].
Case 5 (Scenario: P+ and ↑) EEC in a 3-year-old with primary maxillary anterior teeth with active caries, extraction of the front primary teeth under general anesthesia	17.1%	GA carries risks of major and minor complications and should be avoided if possible [16]. There is no indication for GA as the child is cooperative and treatment may be performed with local anesthesia and behavioral management techniques [19].
Case 5 (Scenario: P+ and ↑) EEC in a 3-year-old with primary maxillary anterior teeth with active caries, strip crown composite restorations	10.4%	Pain without stimulus is a symptom of irreversible pulpitis, where the teeth should not be filled without a pulpotomy/pulpectomy [20].

## Data Availability

The original contributions presented in this study are included in the article. Further inquiries can be directed to the corresponding author.

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
