# Peer review of "Dental Decision-Making in Pediatric Dentistry: A Cross-Sectional Case-Based Questionnaire Among Dentists in Germany"

_medicina, 2024, doi:10.3390/medicina60111907_

Round 1

Reviewer 1 Report

Comments and Suggestions for Authors

This is a research based on a questionnaire about how is the management of caries according to the guidelines of minimally invasive dentistry in pediatric dentistry in a group of dentists from Germany. The topic is a new one in pediatric dentistry. Furthermore, as the authors explain, this is the first research in Germany to explore dental decision-making in pediatric dentistry.

I have some comments about the paper:

Introduction

Line 79: I partially agree with this sentence, because these treatments are the easiest for non cooperative patients, but could be applied in different types of patients: cooperative and non cooperative children. I suugest take it into account for future studies.

Materials and methods:

Study population: I recommend for future investigations to make a more homogeneus sample, only with experienced pediatric dentists, with a widely professional career.

On the other hand, for possible future studies, I would suggest adding X-rays as a complementary test to cases 1 and 2, even though the colour of the caries clearly indicates that the lesions are chronic and the others active, respectively.

For the rest of the paragraphs, I have no comments to add. I found it a really interesting survey to find out whether and under what conditions minimally invasive dentistry criteria are really being applied among pediatric dentists. Based on the results obtained, many university professors will also be able to adapt their teaching so that students better understand this concept, even though its application depends on other additional circumstances, as detailed in the manuscript.

Author Response

 Comment 1

This is a research based on a questionnaire about how is the management of caries according to the guidelines of minimally invasive dentistry in pediatric dentistry in a group of dentists from Germany. The topic is a new one in pediatric dentistry. Furthermore, as the authors explain, this is the first research in Germany to explore dental decision-making in pediatric dentistry.

 I have some comments about the paper:

 Introduction

Line 79: I partially agree with this sentence, because these treatments are the easiest for non cooperative patients, but could be applied in different types of patients: cooperative and non cooperative children. I suggest take it into account for future studies.

Response 1: Thank you for your comment; the sentence was accordingly modified.

Comment 2

Materials and methods:

Study population: I recommend for future investigations to make a more homogeneus sample, only with experienced pediatric dentists, with a widely professional career.

Response 2: Yes, thank you, this is a very good point, that we will consider for future studies.

Comment 3

On the other hand, for possible future studies, I would suggest adding X-rays as a complementary test to cases 1 and 2, even though the colour of the caries clearly indicates that the lesions are chronic and the others active, respectively.

Response 3: Thank you, we will consider this for future studies.

Comment 4

For the rest of the paragraphs, I have no comments to add. I found it a really interesting survey to find out whether and under what conditions minimally invasive dentistry criteria are really being applied among pediatric dentists. Based on the results obtained, many university professors will also be able to adapt their teaching so that students better understand this concept, even though its application depends on other additional circumstances, as detailed in the manuscript.

Response 4: Thank you very much for your valuable comments as well as the recommendations for future studies.

Reviewer 2 Report

Comments and Suggestions for Authors

Dear Authors,

Your study, titled " Dental decision-making in pediatric dentistry: a cross-sectional case-based questionnaire among dentists in Germany," addresses a timely and highly relevant topic for the scientific community. The article is well-structured, with ideas clearly and coherently presented, which enhances readers’ understanding of the study. Additionally, the methodology used is appropriate for the study's objectives, with careful attention to detail and data validity, which adds credibility to the results.

The only suggested improvements would be the inclusion of references to support the ideas presented in lines 45-49 and 398-400.

Overall, I find the article to be well-executed and requiring no further modifications.

Sincerely,

Author Response

Comment 1

Dear Authors,

Your study, titled " Dental decision-making in pediatric dentistry: a cross-sectional case-based questionnaire among dentists in Germany," addresses a timely and highly relevant topic for the scientific community. The article is well-structured, with ideas clearly and coherently presented, which enhances readers’ understanding of the study. Additionally, the methodology used is appropriate for the study's objectives, with careful attention to detail and data validity, which adds credibility to the results.

The only suggested improvements would be the inclusion of references to support the ideas presented in lines 45-49 and 398-400.

Overall, I find the article to be well-executed and requiring no further modifications.

Sincerely,

Response 1: Thank you very much for your comments and your review! The citations were added according to your recommendation. Your comments and compliments are highly appreciated. Thank you!